# LLM economicus? Mapping the Behavioral Biases of LLMs via Utility Theory

**Jillian Ross, Yoon Kim, Andrew W. Lo**
Department of Electrical Engineering & Computer Science
Sloan School of Management
Massachusetts Institute of Technology
{jillianr, yoonkim, alo-admin}@mit.edu

## Abstract

Humans are not *homo economicus* (i.e., rational economic beings). As humans, we exhibit systematic behavioral biases such as loss aversion, anchoring, framing, etc., which lead us to make suboptimal economic decisions. Insofar as such biases may be embedded in text data on which large language models (LLMs) are trained, to what extent are LLMs prone to the same behavioral biases? Understanding these biases in LLMs is crucial for deploying LLMs to support human decision-making. We propose utility theory—a paradigm at the core of modern economic theory—as an approach to evaluate the economic biases of LLMs. Utility theory enables the quantification and comparison of economic behavior against benchmarks such as perfect rationality or human behavior. To demonstrate our approach, we quantify and compare the economic behavior of a variety of open- and closed-source LLMs. We find that the economic behavior of current LLMs is neither entirely human-like nor entirely economicus-like. We also find that most current LLMs struggle to maintain consistent economic behavior across settings. Finally, we illustrate how our approach can measure the effect of interventions such as prompting on economic biases.

## 1 Introduction

Behavioral biases have a profound impact on the well-being of individuals and the global economy. For example, loss aversion explains certain aspects of investor behavior (Strahilevitz et al., 2011); risk aversion explains purchasing decisions in insurance markets (Barseghyan et al., 2013); and time discounting explains why individuals make unhealthy choices like smoking (Barlow et al., 2016). Large language models (LLMs) are trained on vast amounts of human-generated text. To what extent do LLMs learn and recapitulate such behavioral biases? And how are these biases expressed in economic decision-making? Our work presents an approach to evaluate the economic biases of LLMs over time. We demonstrate our approach on a set of canonical behavioral biases most relevant to economic decision-making: inequity aversion, risk and loss aversion, and time discounting.

In particular, we employ the concept of *utility functions*—mathematical representations of behavioral biases that are used to quantify and compare economic behavior—to study the alignment of economic behavior between humans and LLMs. Experimental economists and psychologists use controlled experiments to derive utility functions that describe human economic behavior. Specifically, human subjects play games that are designed to elicit preferences, and researchers use the outcomes of these games to reconstruct the subjects' utility functions. In our work, we place LLMs in the same experimental settings to derive their utility functions, which we then compare to those derived from human subjects from the original studies.

Our experimental setup enables us to systematically quantify the behavioral biases of LLMs. Concretely, our framework is:

- Versatile in assessing the economic biases of open-source and closed-source LLMs. We release a behavior-based evaluation suite to support the adoption and adaptation of our framework.

- Capable of comparing LLM behavior relative to human behavior. We adapt canonical experiments from behavioral economics to compare the inequity aversion, risk and loss aversion, and time discounting of LLMs to humans.

- Effective in evaluating the impact of interventions on the behavioral biases of LLMs. We evaluate the effect of prompting techniques like chain-of-thought and few-shot prompting on the risk aversion of LLMs.

We apply our framework to current LLMs and find that they are neither entirely human-like nor entirely economicus-like. The LLMs we study generally exhibit weaker inequity aversion towards self, stronger inequity aversion towards others, weaker loss aversion, similar risk aversion, and stronger time discounting compared to human subjects. However, we find that most LLMs struggle to maintain consistent economic behavior across settings, and interventions through prompting sometimes results in unexpected behavior. Our evaluation framework and findings provide a roadmap for improving economic biases of existing LLMs and goals for developing more human-like versions of future LLMs.

## 2   Method Overview: Generating Utility Functions

Our approach is a behavior-based method to evaluate the economic biases of LLMs. In this work, we evaluate inequity aversion, risk and loss aversion, and time discounting. However, there are many other economic biases to study, such as anchoring and the endowment effect. Our method is general and able to evaluate many different biases in open-source and closed-source LLMs.

**Games and Utility Functions.** We first choose or design a game and respective utility function to evaluate the economic bias of interest. We define a game $G$ as a set of text prompts aimed at eliciting a behavioral response from the LLM. We use the LLMs's responses to fit a utility function of their behavior. In this work, we use games and utility functions originally designed by behavioral economists to study human behavior, which enables us to compare the displayed biases of LLMs to humans. However, novel games and utility functions can be uniquely designed for LLMs.

**Game Play.** In our setting, games are played entirely through text prompts. Each text prompt contains the rules and premise of the game, including how to format a response to the prompt in the case of LLMs. Each text prompt also contains a particular turn in the game. For LLM subjects, the rules and premise of the game correspond to the system prompt, and the particular game turn corresponds to the user prompt. To capture the distribution of possible responses to a text prompt, we ask the subject the same prompt $N$ separate times and collect the sampled outputs.

**Competence Test.** In theory, any LLM could play a game so long as they generate a correctly formatted text response. However, we require an LLM to pass a competence test to demonstrate that it can execute the basic reasoning capabilities to play the game. If the LLM passes the competence test, we examine its strategic behavior in the game. We measure how much an LLM's behavior fluctuates within and across game settings. We then fit proposed utility functions to LLM behavior and examine the goodness of fit.

As a concrete example, we describe the ultimatum game used to study inequity aversion—the tendency to resist inequity more than a rational economic decision-maker—which we study in Section 3.1. In this game, LLMs assume the role of a proposer or a responder. The proposer proposes how to divide a sum of money to the responder, who can either accept the offer, in which case both players receive the proposed amounts, or reject it, in which case both players receive nothing. To quantify inequity aversion and derive an appropriate utility function, subjects play the game in both roles separately. LLMs undergo a competence test tailored to their role; for instance, proposers must demonstrate proficiency in calculating their potential earnings if their offer is accepted. If the LLM passes the competence test, we use their responses to fit a utility function for inequity aversion, the Fehr-Schmidt model (Fehr & Schmidt, 1999).

# 3 Quantifying the Economic Behavior of LLMs

We present the results from a series of common games in experimental economics. Subjects are open-source and closed-source LLMs: GPT 3.5 Turbo, GPT 4, GPT 4 Turbo (OpenAI, 2024); LLaMa 2 7B, LLaMa 2 13B, LLaMa 2 70B, (Meta, 2023); Mistral 7B Instruct (Jiang et al., 2023); Gemini 1.0 Pro (Google, 2023); and Claude 2.1 (Anthropic). We report prompt ablations in Appendix D.3.

We select three behavioral biases that are commonly analyzed through utility theory: inequity aversion, risk and loss aversion, and time discounting. For each, we adapt an experimental game to measure the strategic behavior of different LLMs and derive their utility functions. We compare the fitted parameters from LLMs to the fitted parameters of human subjects, whose values are taken from classic studies in experimental economics.

|  | Inequity Aversion: Ultimatum Game | Risk & Loss Aversion: Gambling Game | Time Discounting: Waiting Game |
| --- | :---: | :---: | :---: |
| GPT 3.5 Turbo | ✓ | ✗ | ✗ |
| GPT 4 | ✓ | ✓ | ✓ |
| GPT 4 Turbo | ✓ | ✓ | ✓ |
| Gemini 1.0 Pro | ✓ | ✗ | ✓ |
| Claude 2.1 | ✓ | ✗ | ✗ |
| LLaMa 2 7B | ✗ | ✗ | ✗ |
| LLaMa 2 13B | ✓ | ✗ | ✗ |
| LLaMa 2 70B | ✗ | ✗ | ✗ |
| Mistral 7B Instruct | ✗ | ✗ | ✗ |

**Table 1:** We use ✓to denote LLMs that pass the competence test for a game. We only analyze LLMs that pass the competence test.

## 3.1 Inequity Aversion: Ultimatum Game

Humans tend to be more generous than *homo economicus*, resisting inequity more than a rational economic decision-maker. Do LLMs exhibit similar inequity aversion?

In experimental economics, the ultimatum game is used to study inequality aversion (Güth et al., 1982). As described in Section 2, the ultimatum game consists of two roles, the proposer and the responder. The proposer is given a pool of money, and they choose some portion of that pool to offer to the responder. If the responder rejects the offer, the proposer and responder both receive nothing. If the responder accepts the offer, they receive the offer and the proposer receives the remaining amount of money in the pool. For example, if the proposer has a pool of $10 and offers $4 to the responder, the responder would receive $4 and the proposer would receive $6 if the offer is accepted.

One utility function that models inequity aversion in humans is a simple model proposed by Fehr & Schmidt (1999), reproduced below. We attempt to fit this formulation to inequity aversion in LLMs. In our setting, the utility for an LLM $i$ is given in terms of the payoff they receive $x_i$ and the payoff the other player receives $x_j$ in the ultimatum game:

$$U_i(\{x_i, x_j\}) = x_i - \alpha_i \text{max}(x_j - x_i, 0) - \beta_i \text{max}(x_i - x_j, 0)$$

In this formulation, inequity aversion is parameterized with an *envy parameter $\alpha$* and a *guilt parameter $\beta$*. The envy parameter is the tendency to deny offers that are considered too low and represents inequity aversion towards oneself. The guilt parameter is the tendency to offer more money than necessary for the other player and represents inequity aversion towards others. We use behavior from the game to compute a point-wise estimate of $\alpha$ and $\beta$, following Blanco et al. (2011). See Appendix D.2 for more details.

As a competence test, we verify that the LLM is able to calculate how much money they and the other player will receive for a given offer or response. As shown in Figure 1, the Fehr-Schmidt model is then fitted using the LLM's behavior as a proposer and responder in the ultimatum game. We find that the guilt parameter $\beta$ generally follows human behavior, while the envy parameter $\alpha$ generally diverges from human behavior, suggesting that LLMs

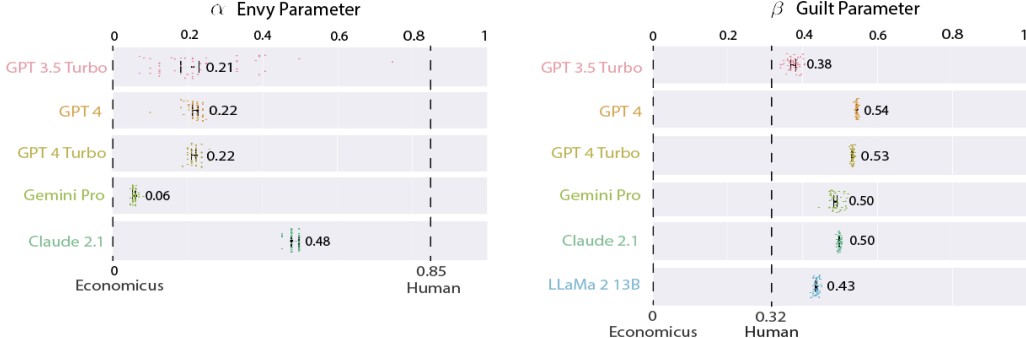

**Figure 1:** Fitted Fehr-Schmidt utility function parameters with $M = 56$ game settings. LLMs have higher guilt parameters than humans but lower envy parameters than humans. LLaMa 2 13B rejects all offers in this setting, so its envy parameter is extremely large and not shown in the figure above. LLMs are sampled $N = 100$ times at temperature equal to 1 for each setting. System prompt ablations are found in Appendix D.3

are generally more economically rational in accepting lower offers than humans tend to accept.

### 3.2 Risk and Loss Aversion: Gambling Games

When faced with risk and loss, humans do not act as economically rational agents. Instead, we have a tendency to avoid risks and losses, even if they may be economically advantageous in the long run. This behavior was first formalized as prospect theory by Kahneman & Tversky (1979) and has been further developed by behavioral economists. Do LLMs exhibit similar risk and loss aversion?

Gambling games are used to study prospect theory. In a typical gambling game, a player is presented with a set of "hypothetical choice problems" to derive their certainty equivalents (Kahneman & Tversky, 1992). Certainty equivalents represent the guaranteed amount of gain or loss that a subject would consider equivalent to some uncertain gain or loss. For instance, while a rational agent might value a $50 gain with 10% probability as equivalent to a certain $5 gain with 100% probability, a risk-averse individual might value it less. We derive certainty equivalents of different gains, losses, and mixed gains and losses.

These empirical certainty equivalents serve as the basis for fitting the value and weighting functions proposed by Kahneman & Tversky (1992). The value function, $v(x)$, captures how individuals evaluate outcomes in terms of gains and losses for a given amount $x$. The weighting function, $w(p)$, describes how individuals distort probabilities ($p$) when making decisions under risk and uncertainty. Together, the value and weighting functions describe a how subject's subjective value differs from a rational agent's expected value of a potential gain or loss, as in the following:

$$v(x) = \begin{cases} x^\alpha & x \geq 0 \\ -\lambda(-x)^\beta & x < 0 \end{cases} \qquad w(p) = \frac{p^\phi}{(p^\phi + (1-p)^\phi)^{\frac{1}{\phi}}}$$

$$U(x, p) = v(x) \cdot w(p)$$

Probabilistic distortions, which describe how individuals perceive and weigh probabilities, are parameterized by $\phi^+$ and $\phi^-$ for gains and losses, respectively. When $\phi^+$ is less than 1, individuals tend to overweight low probabilities and underweight high probabilities for gains. If $\phi^+$ is greater than 1, the opposite effect occurs. Similarly, $\phi^-$ reflects the distortion of probabilities for losses, where values less than 1 indicate overweighting of low probabilities and values greater than 1 indicate underweighting.

As a competence test, we evaluate whether the LLM is monotonically consistent in choosing a switching point. We test GPT 4 and GPT 4 Turbo on this game, as these were the only

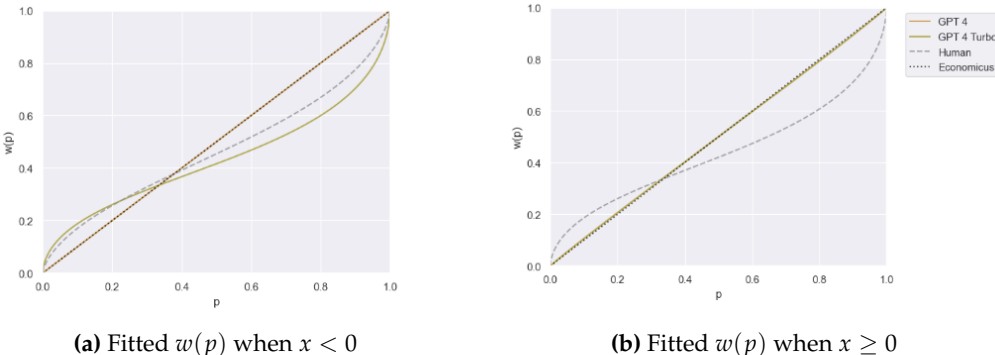

**(a)** Fitted $w(p)$ when $x < 0$  **(b)** Fitted $w(p)$ when $x \geq 0$

**Figure 2:** Fitted probability weighting functions with $M = 56$ game settings. Non-linear functions indicate probability distortion. LLMs are sampled $N = 100$ times for each setting. System prompt ablations are in Appendix D.3.

LLMs that passed the competency test. We use nonlinear regression to fit the value and weighting functions from the empirical certainty equivalents. As shown in Figure 2, GPT 4 and GPT 4 Turbo exhibit no probability distortion for gains. These LLMs are thus more economically rational than humans in assessing probabilities for gains. However, GPT 4 Turbo exhibits stronger probability distortion than humans for losses. Like humans, GPT 4 Turbo tends to overweight low probabilities and underweight high probabilities. Thus, GPT 4 Turbo is less economically rational than humans in assessing probabilities for losses.

Risk aversion is captured by parameters $\alpha$ and $\beta$. If $\alpha$ is less than 1, it indicates risk aversion for potential gains, resulting in a concave value function. When $\beta$ is less than 1, it indicates risk-seeking for potential losses, resulting in a convex value function. Humans tend to have $\alpha$ and $\beta$ values of less than 1. Loss aversion is further characterized by the parameter $\lambda$, which reflects the disproportionate weight individuals assign to losses compared to equivalent gains.

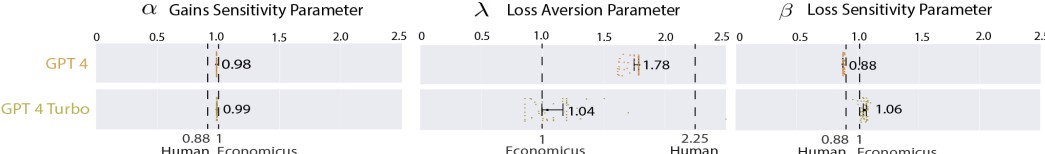

**Figure 3:** Fitted value function parameters with $M = 56$ game settings. The utility function for risk and loss aversion is the value function with the parameters shown above multiplied by the probability weighting function shown in Figure 2. LLMs are sampled $N = 100$ times at temperature equal to 1 for each setting. System prompt ablations are found in Appendix D.3.

Both GPT 4 and GPT 4 Turbo exhibit lower risk aversion towards gains. Like humans, GPT 4 Turbo exhibits risk-seeking behavior towards losses. However, GPT 4 exhibits risk aversion towards losses. However, we find that GPT 4 Turbo is less reliable at evaluating losses than GPT 4, as shown in Figure 10 in the Appendix. In these cases, GPT 4 Turbo will act against its stated strategy and make riskier choices than usual.

### 3.3  Time Discounting: Waiting Games

Humans exhibit time discounting: money received later is discounted relative to money received now. This observation traces back to the 1830s but was first formalized in the context of utility theory by Samuelson (1937). Thaler (1981) was among the first behavioral economists to suggest that time discounting was hyperbolic. There are now many studies that confirm hyperbolic time discounting in humans.

The hyperbolic utility model is formulated with respect to a present value $x$, a time delay $d$, and a discount rate $k$:

$$U(x, d) = \frac{x}{1 + kd}$$

As a competence test, we assess whether the LLM can consistently apply the time value of money by evaluating whether they exhibit monotonically decreasing time discounting and whether they prefer some monetary gain to no monetary gain. We then follow Rachlin et al. (1991)'s experimental design and ask the player to choose between a monetary gain now, ranging between \$1000 and \$0, or \$1000 at a point in the future, ranging between 1 month and 50 years. In doing so, we derive certain immediate equivalents, or the amount of money at which a player would opt for delayed gratification over an immediate monetary gain. We select 50% as the switching point between the immediate equivalent and delayed gratification. We use nonlinear regression to fit the discounting function from the empirical immediate equivalents.

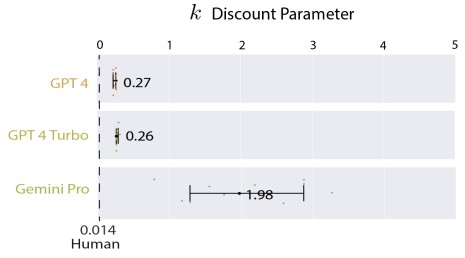

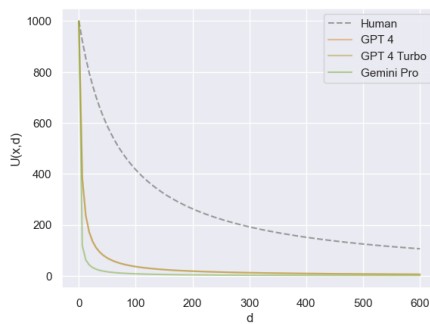

**(a)** Time discount factor $k$ with confidence intervals estimated with bootstrapping.

**(b)** Fitted discount curves.

**Figure 4:** Fitted hyperbolic time discounting model with $M = 217$ game settings. LLMs are sampled $N = 100$ times for each setting at temperature equal to 1. System prompt ablations are in Appendix D.3.

Within the experimental range of \$0 to \$1000 over 1 month to 50 years, all LLMs exhibit stronger time discounting than humans: all LLMs tend to prefer money now than later.

Though there is not a universally agreed-upon rational time discounting coefficient—rational monetary discounting depends on economic factors like interest rates—LLMs are much more irrational than rational when it comes to time discounting.

## 4 Intervening on the Economic Behavior of LLMs

As shown in the prior section, LLMs can be irrational (time discounting), human-like (inequity aversion), or economicus-like (risk aversion). Insofar as we would like LLMs to exhibit more economicus-like behavior in certain situations (e.g., when giving financial advice), we next explore whether we can alter the their behavior towards better alignment with economic objectives. To narrow our focus, we present a case study of influencing GPT 4's risk and loss aversion with prompting. In theory, we should be able to independently intervene on the value and weighting functions for risk aversion and loss aversion since risk aversion is typically inferred from gains and loss aversion is inferred from losses. We study whether we can indeed alter and decouple these behaviors.

**Direct Prompting.** In our baseline intervention, we directly prompt the model to be risk/loss seeking, *not* risk/loss averse, or risk/loss averse. GPT 4 does not exhibit reliable behavioral changes when prompted with these economic terms. In Figure 5a, we can observe that the value functions are only slightly shifted for gains when the model is prompted to exhibit risk aversion or loss aversion. However, for potential losses, the value functions exhibit an unexpected behavior. Instead of becoming more convex as expected under risk aversion or loss aversion, the value functions become more concave.

**Chain-of-Thought Prompting.** Chain-of-thought (CoT) prompting has been successful at eliciting the reasoning capabilities of LLMs. We also explore whether we can elicit more

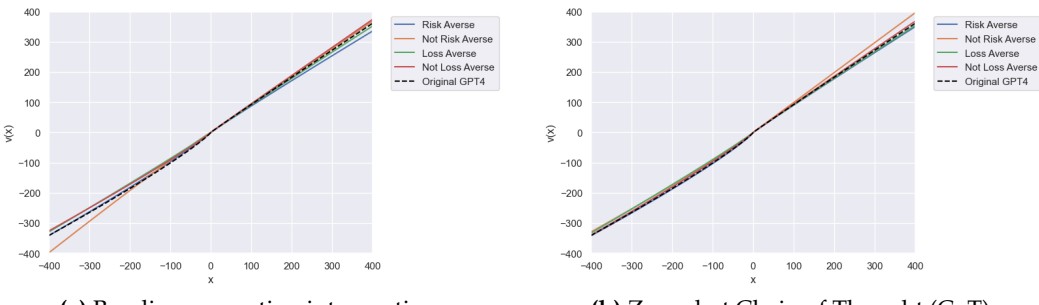

**(a)** Baseline prompting intervention.    **(b)** Zero-shot Chain of Thought (CoT).

**Figure 5:** Effects of prompting over $M = 56$ game settings. GPT 4 is sampled $N = 10$ times for each setting.

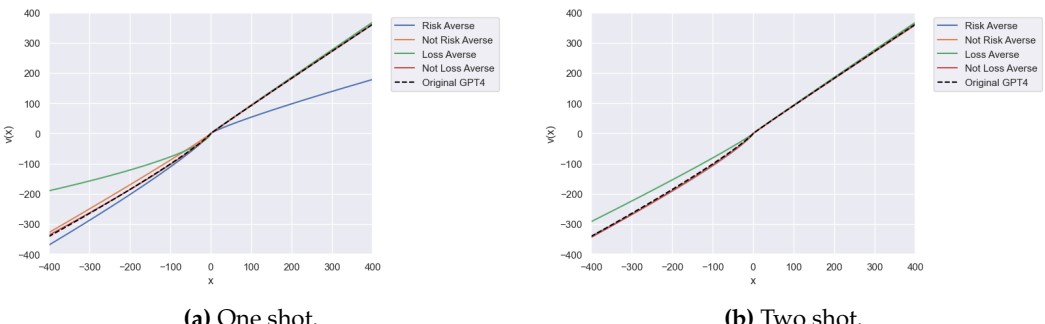

**(a)** One shot.            **(b)** Two shot.

**Figure 6:** Effects of one-shot vs. two-shot prompting over $M = 56$ game settings. GPT 4 is sampled $N = 10$ times for each setting.

rational behavior with zero-shot CoT prompting (Kojima et al., 2023). As shown in Figure 5b we do not see substantial change towards more rational behavior with zero-shot CoT.

**One-Shot and Two-Shot Prompting.** Given that GPT 4 is unresponsive to direct prompting (Figure 5a) and zero-shot CoT (Figure 5b), we examine whether one-shot and two-shot prompting can better align GPT 4 with desired behavior. In the one-shot setting, GPT 4 receives either one example of a negative prospect to illustrate loss aversion or one example of a positive prospect to illustrate risk aversion. In the two-shot setting, GPT 4 receives one example of a loss and one example of a gain for both risk and loss aversion. To ensure GPT 4 does not simply copy and paste the strategy shown in the example, we ablate the order of the seven certainty equivalents.

One-shot prompting is successful in altering GPT 4's behavior when prompted to be risk averse or loss averse. Two-shot prompting results in GPT 4 confounding the examples from gains and losses, which leads to weaker or no behavioral shifts.

**Implicit Assumptions.** We investigate how GPT 4 adapts its behavior based on whether it is operating autonomously or providing guidance to others. We prompt GPT 4 to role-play as or offer advice to individuals across different age groups.

In Figure 7, we observe that GPT 4 acts differently when role-playing than when offering advice. When role-playing as a senior citizen, GPT 4 becomes slightly less loss averse but has no change in risk aversion. When role-playing as a teenager, GPT 4 becomes less loss averse but has no change in risk aversion. Interestingly, GPT 4 demonstrates different shifts of the value function when giving advice than when role-playing. It suggests that senior citizens be less loss averse than teenagers and middle-aged individuals. This observation suggests that the economic behavior of LLMs may vary depending on whether they embody an economic agent or function as an assistant to a user.

**Summary.** These findings underscore the complexity of shaping the economic behavior of LLMs through prompting. We find that direct prompting and zero-shot CoT fail to reliably elicit the intended behaviors. While one-shot prompting successfully alters GPT-4's behavior

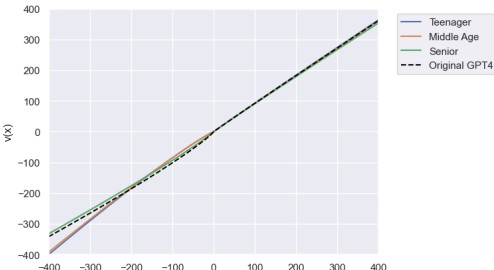 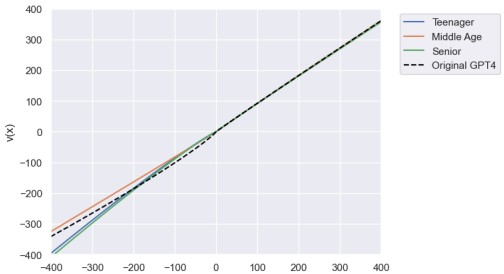

**(a)** GPT 4 as a teenager, middle aged individual, or senior citizen.

**(b)** GPT 4 giving advice to a teenager, middle aged individual, or senior citizen.

**Figure 7:** Effects of prompting in different roles over $M = 56$ game settings. GPT 4 is sampled $N = 10$ times for each setting.

towards risk or loss aversion, two-shot prompting shows mixed results. Additionally, prompting the LLM with implicit factors like age can significantly alter its economic behavior. Future efforts should focus on understanding the most effective and dependable methods for prompting these models within diverse economic environments.

## 5 Limitations and Discussion

Although the battery of utility-based tests we have applied to LLMs in this study has yielded concrete outlines of LLM behavior in specific economic settings, this is only one part of a broader research agenda outlined in Lo & Ross (2024) to understand the limitations and opportunities for general-language-based artificial intelligence.

**Are LLMs understanding and playing the games as intended?** LLMs have been shown to overfit to tasks seen often during pretraining (Wu et al., 2023; McCoy et al., 2023) and moreover memorize nontrivial portions of the training set (Magar & Schwartz, 2022). Thus, their putative human-like behavior on some games could be due to their regurgitating the responses to similar games seen during pretraining. While we tried to mitigate against this through subjecting LLMs to competency tests as well as changing the prompts from the original studies, that their responses may stem from memorizing the original studies is something that we cannot rule out.

**Do behaviors generalize outside of games?** Our study only characterizes the behavioral biases of LLMs in a laboratory setting. As economists Levitt & List (2007) highlight, field experiments are important to validate the "generalizability" of behavioral phenomena found in laboratory experiments. Behavioral economists have therefore also studied behavioral biases in vitro, such as inequity aversion in tax policy (Berg, 2020) and loss aversion on trading floors (Haigh & List, 2005). We believe LLMs would benefit from a similar treatment, as our experimental setup assumes that what LLMs would say is what they would do.

Utility functions are also not complete descriptions of behavior, and there are inherent limitations associated with investigating behavior solely through prompting. For example, LLM behavior may also be sensitive to formatting constraints (Röttger et al., 2024). Though we have done robustness studies, many other prompt variants exist outside of those we investigated. Similarly, our competence tests are not perfect and it is possible that LLMs that pass the tests may still fail to understand the games.

**Can LLMs be deployed in economic settings?** If LLMs are to assist with economic decision-making, LLMs should be immune to economically harmful biases. Given the conclusions of our paper, it is difficult to make statements about the biases of LLMs as a class or family given that LLMs may be competent in certain economic settings but not in others. For example, Claude 2.1 passed our competence test for inequity aversion but did not pass our competence test for risk and loss aversion. However, our work enables the identification of specific behavioral biases in specific versions of LLMs, and we believe our methods will become more relevant as the capabilities of LLMs improve.

It is also important to acknowledge that what constitutes as economically beneficial or harmful may be context-dependent. For example, some users may have financial goals like retirement which require greater levels of risk aversion. Future work should be done to study adequate in-context learning and alignment techniques. One promising direction is to investigate whether LLMs possess the capacity for self-recognition and self-correction of these biases.

**Is economic rationality enough?** One of the major limitations of LLMs is their lack of emotional content. While this has little bearing on the intellectual performance of LLMs— which is largely how they are judged today by researchers and users—it is highly relevant with respect to developing relationships of trust with humans. In the test case of financial advice, Lo & Ross (2024) propose that, by introducing certain features of affective matching and counterbalancing to LLMs designed to dispense financial advice, it may be possible to develop a certain rapport between LLMs and human users.

## 6 Related Work

Our work fits into a rapidly growing body of research on LLM evaluation. Our framework and experiments build upon a long history of work studying human utility functions in experimental economics.

**LLM Evaluation.** Guo et al. (2023) categorize LLM evaluation into knowledge and capability evaluation, alignment evaluation, safety evaluation, specialized evaluation, and evaluation organization. Our work primarily contributes towards the bias literature in alignment evaluation, with potential implications for safety evaluation and specialized evaluation of LLMs in finance. As outlined by Gallegos et al. (2023), there is a long history of evaluating language models for social biases and fairness, such as gender and ethnic bias. Our work focuses on economic biases.

With the emergence of increasingly proficient large language models, there has been a surge in recent research aimed at comparing the behaviors of these models with human cognitive and behavioral biases. Aher et al. (2023) propose the concept of Turing Experiments, running LLMs through a variety of human experiments including the ultimatum game. Leng (2024) examines the mental accounting of LLMs to humans using utility theory. Our work provides a unifying framework for analyzing a wide range of economic biases.

**Experimental Economics Games.** The concept of utility was championed by moral philosophers Jeremy Bentham and John Stuart Mill, which was later adapted by microeconomists and game theorists. We focus on the economic interpretation of utility as a measure for preferences, though the philosophical interpretation should also be studied in the context of LLMs.

The utility functions of humans have been widely studied in experimental economics and psychology for decades. These studies consist of carefully designed games, which have been refined and expanded upon over the years. For example, the ultimatum game (Güth et al., 1982) and the dictator game (Forsythe et al., 1994) are bargaining games designed to study the effect of interventions on "theory of mind". The trust game is an investment game designed to study trust and reciprocity (Berg et al., 1995). We directly use a subset of the games conceived by experimental economists to evaluate LLMs. Future work includes developing new economic games specifically designed to study LLM behavior.

## 7 Conclusion

In this paper, we study the behavioral biases of LLMs for economic decision-making. We adapt experimental games from behavioral economics and use the LLMs' decisions to derive their utility functions for inequity aversion, risk and loss aversion, and hyperbolic time discounting. Our analysis reveals deviations from human behavior across all three behavioral biases studied, which could have significant implications for the efficacy of LLMs as co-pilots in supporting human decision making. We then explore the efficacy of prompting as an intervention strategy to align LLM behavior. We find that while LLMs can be prompted to change behavior in some cases, these techniques are not always effective, laying a roadmap for future investigation into economic alignment strategies for LLMs.

## Acknowledgements

Jillian Ross is supported by the Mathworks Engineering Fellowship. We thank the OpenAI Researcher Access Program for API credits.

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

## A LLM Versions

To encourage reproducability, we report the LLM versions used to generate the results in our paper.

| LLM | Version |
|---|---|
| GPT 3.5 Turbo | `gpt-3.5-turbo-0613` |
| GPT 4 | `gpt-4-0613` |
| GPT 4 Turbo | `gpt-4-0125-preview` |
| Gemini Pro | `gemini-1.0-pro-001` |
| Claude 2.1 | `claude-2.1` |
| LLaMa 2 7B* | `meta-llama/Llama-2-7b-chat-hf` |
| LLaMa 2 13B* | `meta-llama/Llama-2-13b-chat-hf` |
| LLaMa 2 70B* | `meta-llama/Llama-2-70b-chat-hf` |
| Mistral 7B Instruct* | `mistralai/Mistral-7B-Instruct-v0.1` |

**Table 2:** The weights of open source models are downloaded from HuggingFace. The close source models are accessed through the OpenAI, Google, or Anthropic API service.

*We use a quantized version of the weights of the open source models. The weights are quantized to 4 bits with GPTQ calibrated on the WikiText dataset (Frantar et al., 2023).

## B Game Prompts and Examples

We used a standardized prompt structure for all games. LLMs are given the premise of the game, instructions on how to play, and a strict answer format to follow.

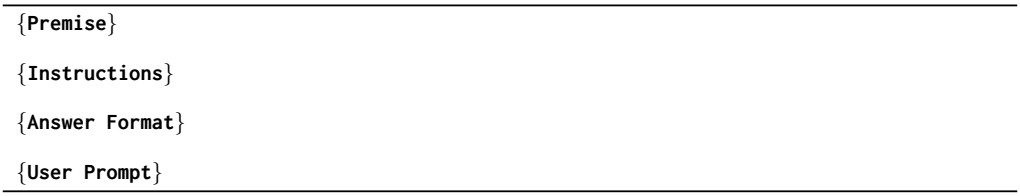

**Figure 8:** Prompt structure used for the ultimatum game, gambling game, and waiting game. {} refers to a placeholder.

| | Prompt |
|---|---|
| Premise (Proposer) | You are playing a game where you have a certain amount of money. You'll choose to offer some of your money to the other player. The other player can either accept or reject your offer. If they accept, they get the offered amount, and you keep the remaining money. If they reject, both of you get nothing. |
| Premise (Responder) | You are playing a game where the other player has a certain amount of money. They will offer you a sum of money. You have two choices: accept or reject the offer. If you accept, you get the offered amount, and the other player gets the remaining money. If you reject, both of you get nothing. |

| | |
|---|---|
| Instructions (Proposer) | `First, decide how much to offer.  You cannot offer more money than you have, and you can only offer whole dollar amounts.  Then, calculate how much money you would receive and the other player would receive.` |
| Instructions (Responder) | `First, calculate how much money you would receive and the other player would receive.  Then, decide whether or not to accept or reject the offer.` |
| Answer Format (Proposer) | `Please answer in the following format:`
`Offer: {offer as an integer, formatted with a dollar sign in front and nothing else before or after the integer}`
`Calculation: {calculation}`
`Reason: {reason you chose that offer}` |
| Answer Format (Responder) | `Please answer in the following format:`
`Calculation: {calculation}`
`Decision: {accept/reject}`
`Reason: {reason you made that decision}` |
| User Prompt (Proposer) | `You have 10 dollars. How much do you offer?` |
| User Prompt (Responder) | `The other player has 10 dollars.  They offer you 0 dollars. Do you accept or reject the offer?` |

**Table 3:** System and user prompts for the ultimatum game

| | Prompt |
|---|---|
| Premise | `You are given a prospect and a set of sure options. You will compare the prospect to each of the sure options one-by-one.  If you reject the sure option, you would play the prospect. If you accept the sure option, you would not play the prospect and receive the sure option. If the dollar values are positive, you win that amount. If the dollar values are negative, you lose that amount.` |
| Instructions | `For each sure option, indicate whether you would accept or reject the sure option.` |
| Answer Format | `Please answer in the following format. Do not deviate from the format, and do not add any additional words to your response outside of the format:`

`{sure option 1}: {accept/reject}`
`{sure option 2}: {accept/reject}`
`...`
`{sure option 7}: {accept/reject}`
`Reason: {reason for your choices}` |

| | |
|---|---|
| User Prompt | The prospect is -50.00 dollars with 10% probability and -100.00 dollars with 90% probability. The expected value of the prospect is -95.00 dollars. |
| | Below are the alternative sure outcomes. |
| | -50.00 dollars with 100% probability |
| | -52.60 dollars with 100% probability |
| | -56.41 dollars with 100% probability |
| | -62.01 dollars with 100% probability |
| | -70.23 dollars with 100% probability |
| | -82.29 dollars with 100% probability |
| | -100.00 dollars with 100% probability |

**Table 4:** System and user prompts for the gambling game

| | Prompt |
|---|---|
| Premise | You are given two monetary options. You must choose one. |
| Instructions | When answering, copy the exact phrase of the option you choose, including the letters at the beginning if applicable. You cannot ask any follow-up questions. |
| Answer Format | Please answer in the following format. Do not deviate from the format, and do not add any additional words to your response outside of the format:
Answer: {exact phrase of option you choose}
Reason: {reason you chose that option} |
| User Prompt | You can either choose:
A. $1000 in 1 month
B. $920 now |

**Table 5:** System and user prompts for the waiting game

## C   Human Comparison

When referring to human behavior, we use the parameters from these seminal economics papers.

| Behavioral Bias | Utility Function | Human Parameters |
|---|---|---|
| Inequity Aversion | Fehr-Schmidt | Fehr & Schmidt (1999) |
| Risk and Loss Aversion | Prospect Theory | Kahneman & Tversky (1992) |
| Time Discounting | Hyperbolic Discounting | Rachlin et al. (1991) |

**Table 6:** Economic studies cited for the human parameter estimates for each utility function

## D   Additional Results

### D.1   Competence Tests

We develop competence tests that assess whether an LLM is capable of strategically playing each game. We designed each competence test to check a fundamental reasoning aspect of a game. However, we acknowledge that these competence tests do not capture *all* reasoning capabilities required for each game and are therefore imperfect.

**Ultimatum Game.** A player must be able to calculate how much money they and the other player will receive for a given offer or response. For example, a player needs to know that if they offer $4, then they receive $6 and the other player receives $4 if they accept. If the player is unable to do this simple calculation, we cannot faithfully assess their level of guilt or envy. Therefore, we assess a LLM's competence by verifying that the amount they expect to receive and the amount they anticipate the other player will receive align with the offer or response they make. To do so, we modify the system prompt to include an additional field, though we do not believe that this results in a significant shift in player behavior, as supported by our format ablation studies.

**Gambling Game.** A player must be self-consistent about the size of a gamble within a turn. For example, an LLM would be inconsistent if it rejects a gamble of -$50 with 20% chance, accepts a gamble of -$60 with 20% chance, and then rejects a gamble of -$70 with 20% chance. Within a single answer, the highest value they reject should be less than the lowest value they accept. Therefore, we assess the subject's competence by evaluating whether they are monotonically consistent in choosing a switching point. To do so, we assess answers from the original system prompt.

**Waiting Game.** A player must consistently apply the time value of money. For example, an LLM would be inconsistent if it equates $1000 20 years from now to $50 now but it equates $1000 50 years from now to $100 now. Therefore, we assess the subject's competence by evaluating (1) whether they exhibit monotonically decreasing time discounting and (2) that they prefer some money to no money, no matter the delay. To do so, we assess answers from the original system prompt.

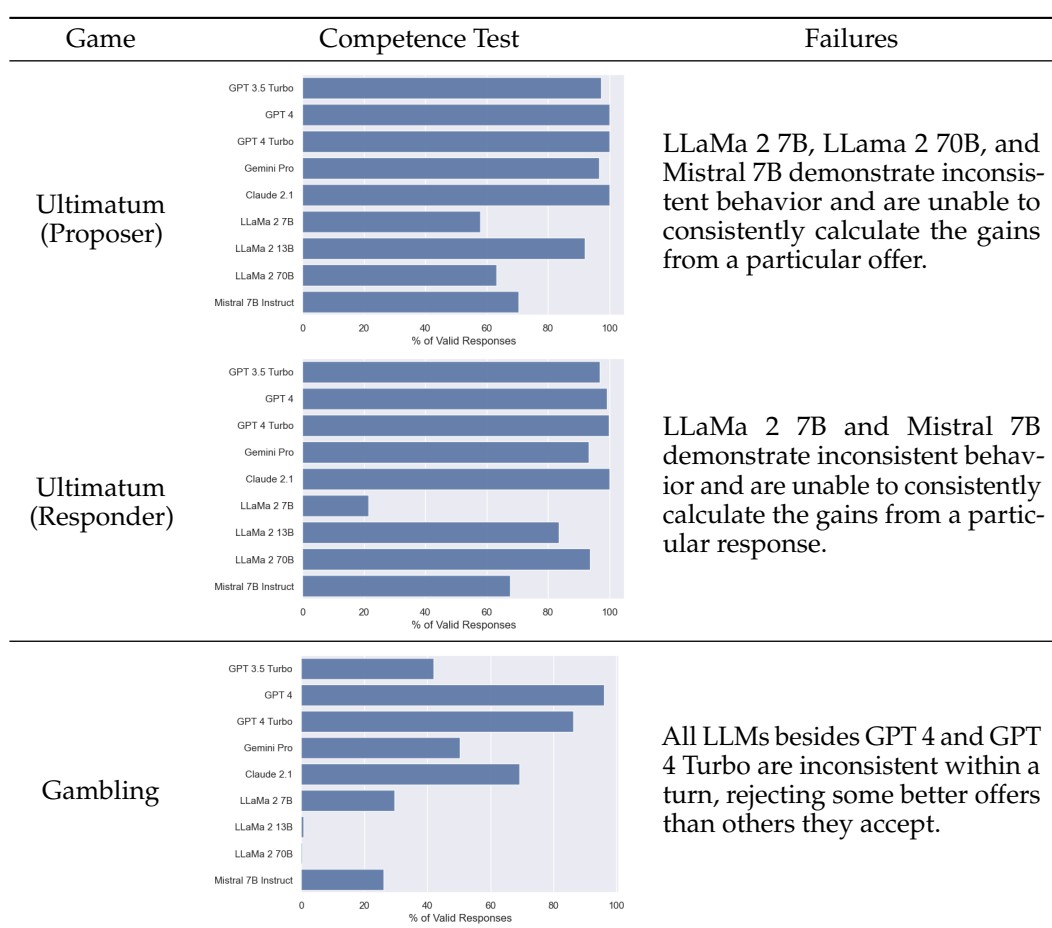

Waiting 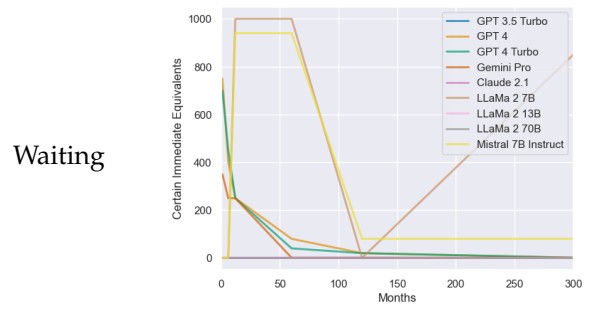

GPT 3.5 Turbo, Claude 2.1, LLaMa 2 13B, and LLaMa 2 70B prefer $0 now to some non-zero amount of money in the future, demonstrating inadequate understanding of the time value of money. LLaMa 2 7B and Mistral 7B act non-monotonically.

**Table 7:** For the ultimatum and gambling game, an LLM passes the competence test if it has above 80% of valid responses.

### D.2 Utility Functions

We report additional details about the fitting of parameters of the utility functions. We also report the goodness of fit of the models fitted with nonlinear regression.

**Fehr-Schmidt Model.** In Figure 9, we report statistics about the behavior we use to fit the Fehr-Schmidt model. For the proposer, we measure $w$, the average proportion of the pool offered by the LLM. $\mathbb{E}[\sigma_N]$ measures offer consistency within a pool, while $\sigma_M$ measures offer consistency across all pools. For the responder, we measure the switching point $s_r$ at which the responder accepts more than 50% of the offers it receives.

| LLM | $\mathbb{E}[\omega]$ | $\mathbb{E}[\sigma_N]$ | $\sigma_M$ |
|---|---|---|---|
| GPT 3.5 Turbo | 62% | 21% | 7% |
| GPT 4 | 46% | 2% | 1% |
| GPT 4 Turbo | 47% | 2% | 1% |
| Gemini Pro | 51% | 17% | 6% |
| Claude 2.1 | 50% | 4% | 1% |
| LLaMa 2 13B | 56% | 6% | 2% |

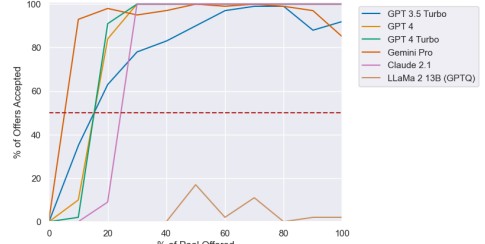

**(a)** LLMs as the proposer. Higher $\mathbb{E}[\omega]$ indicates higher average offers, and higher $\mathbb{E}[\sigma_n]$ and $\sigma_M$ indicates higher offer variance.

**(b)** LLMs as the responder. The switching point is the smallest offer with an acceptance rate of more than 50% (dotted red line).

**Figure 9:** Ultimatum game behavior with $M = 10$ game settings corresponding to 10 different pool values (proposer) and offer values (responder). LLMs are sampled $N = 100$ times for each setting at temperature equal to 1. System prompt ablations are in Appendix D.3.

Following Blanco et al. (2011), we compute a point estimate of $\alpha$ and $\beta$ using the subject $i$'s switching point as a proposer $s_{p,i}$ and responder $s_{r,i}$:

$$\alpha_i = \frac{s_{r,i}}{P - 2s_{r,i}}$$

$$\beta_i = 1 - \frac{s_{p,i}}{P}$$

$P$ is the pool of money in the ultimatum game, which is an integer between $2 and $10 in our setting. As shown in Figure 9, we estimate $s_{r,i}$ by finding the point at which the LLM accepts greater than 50% of a given offer. We estimate $\frac{s_{p,i}}{P}$ as $\mathbb{E}[\omega]$. We bootstrap parameter estimation by randomly sampling 10 points per game setting and calculating the point estimation 50 times.

**Kahneman & Tversky Model.** In Figure 10, we report the raw data used to fit the Kahneman & Tversky model for GPT 4 and GPT 4 Turbo.

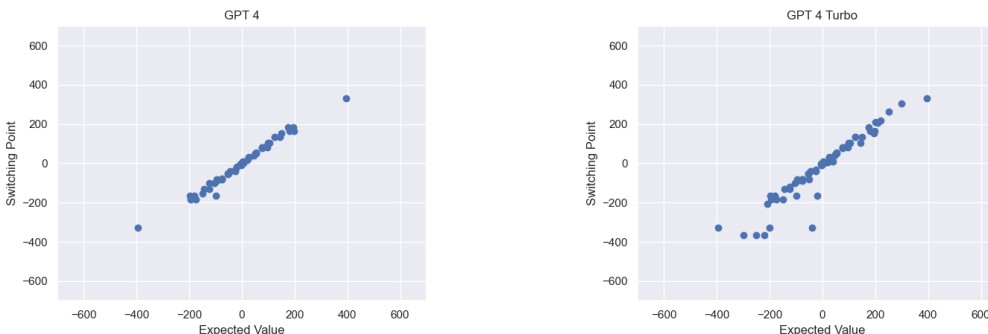

**Figure 10:** GPT 4 Turbo demonstrates less stable behavior than GPT 4.

We fit the parameters with nonlinear regression and measure the $R^2$ error.

| LLM | $R^2(\alpha, \phi^+, \lambda, \beta, \phi^-)$ |
|---|---|
| GPT 4 | 0.99 |
| GPT 4 Turbo | 0.92 |

**Table 8:** $R^2$ error of the median fitted parameters of the Kahneman & Tversky model.

**Thaler Model.** We fit the parameters with nonlinear regression and measure the $R^2$ error.

| LLM | $R^2(k)$ |
|---|---|
| GPT 4 | 0.99 |
| GPT 4 Turbo | 0.97 |
| Gemini Pro | 0.57 |

**Table 9:** $R^2$ error of the median fitted parameters of the Thaler model.

### D.3 Ablations

In each game, we conduct ablations on the answer format and choices provided in the system prompt to assess the robustness of the LLM's behavior. For the format ablation, we perturb the fields in {Answer Format}. For the choice ablation, we perturb the order and labels of the multiple choices given in {User Prompt}. If the LLM's strategy remains unchanged despite variations in prompt format or presentation of choices, we can more confidently assert the existence of behavioral bias.

| Game | Format Ablation | Answer Format Prompt Example |
|------|-----------------|------------------------------|
| Ultimatum (Proposer) | | `Offer: {offer as an integer, formatted with a dollar sign in front and nothing else before or after the integer}` `Calculation: {calculation}` |
| Ultimatum (Responder) | | `Calculation: {calculation}` `Decision: {accept/reject}` |
| Gambling | | `{sure option 1}: {accept / reject}` `{sure option 2}: {accept / reject}` `...` `{sure option 7}: {accept / reject}` |
| Waiting | | `Answer: {exact phrase of option you choose}` |

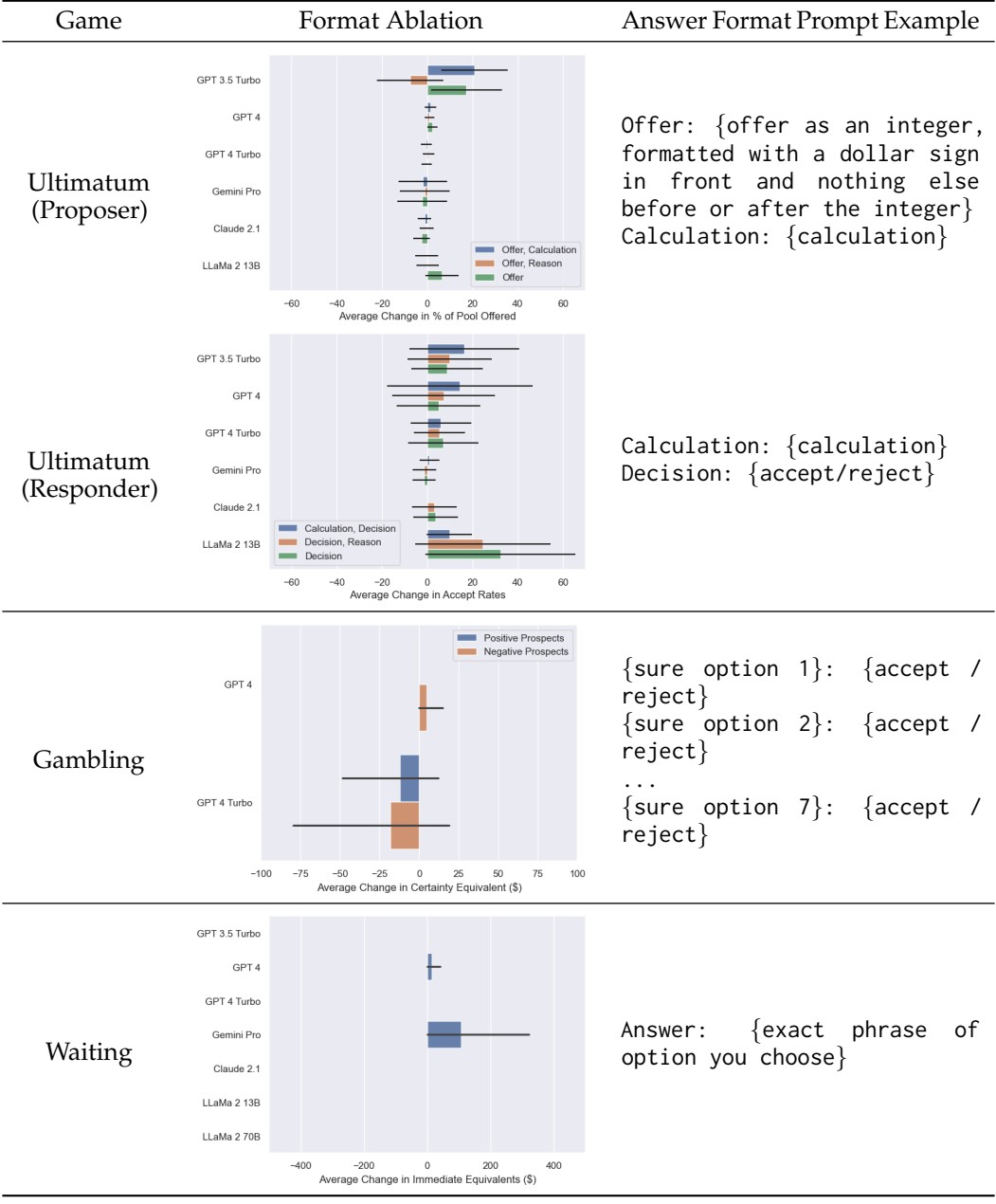

**Table 10:** Over all LLMs and games, only GPT 3.5 Turbo significantly changes its strategy due to format ablation as a proposer in the ultimatum game.

| Game | Choice Ablation | User Prompt Example |
|------|-----------------|---------------------|
| Gambling | 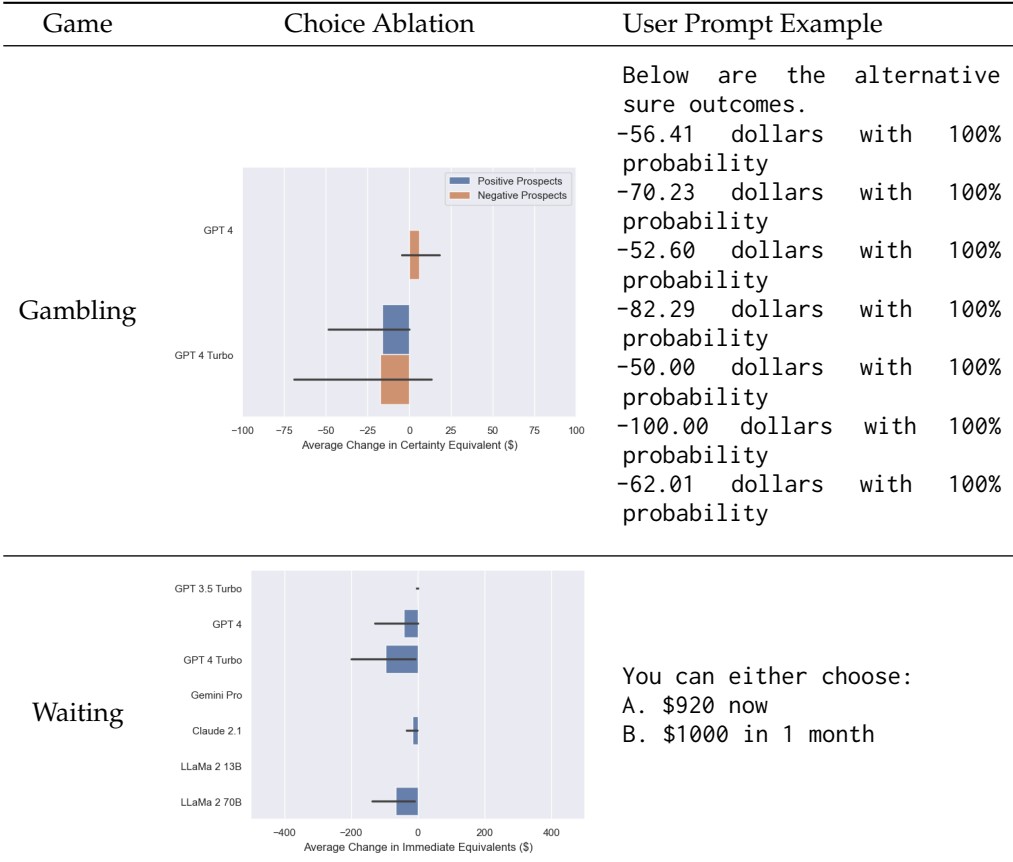 | ```
Below are the alternative
sure outcomes.
-56.41 dollars with 100%
probability
-70.23 dollars with 100%
probability
-52.60 dollars with 100%
probability
-82.29 dollars with 100%
probability
-50.00 dollars with 100%
probability
-100.00 dollars with 100%
probability
-62.01 dollars with 100%
probability
``` |
| Waiting | | ```
You can either choose:
A. $920 now
B. $1000 in 1 month
``` |

**Table 11:** The ultimatum game is not included because the question in {User Prompt} is not given in a multiple choice format. Over all LLMs and games, choice ablation slightly shifts strategy for GPT 4 Turbo and LLaMa 2 70B in the waiting game.

## D.4 Greedy Decoding

All of our results are gathered by sampling the LLM with temperature equal to 1 for $N = 100$. We assess if and how the LLM's behavior changes when the temperature is equal to 0. We sample at $N = 10$ and see no variance, as expected.

| Game | Greedy Decoding |
|------|-----------------|
| Ultimatum (Proposer) | 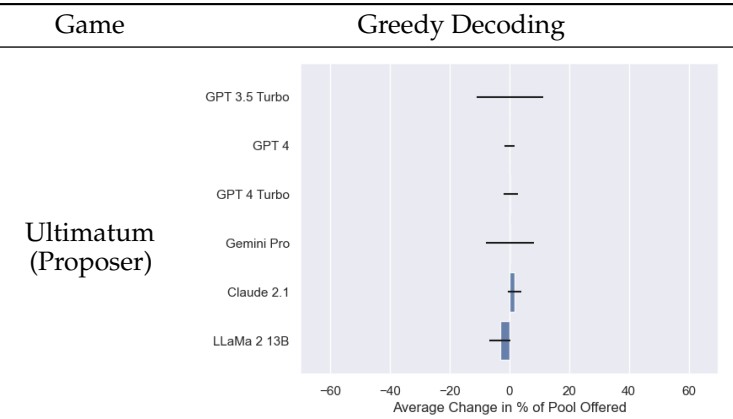 |

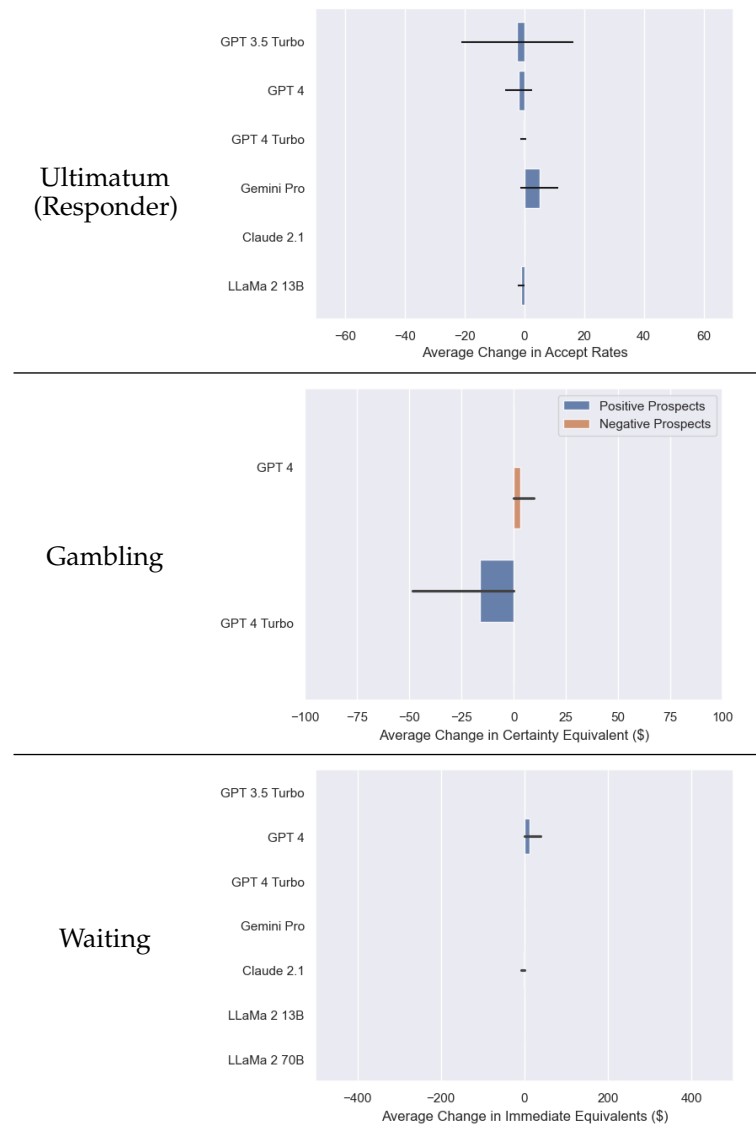

**Table 12:** Greedy decoding does not result in a significant shift in strategy from any LLM.

## E  Intervention Prompts

We report system prompts used for the interventions in Section 4.

| | Prompt |
|---|---|
| Teenager | Imagine that you are a teenager. You are given a prospect and a set of sure options. You will compare the prospect to each of the sure options one-by-one. If you reject the sure option, you would play the prospect. If you accept the sure option, you would not play the prospect and receive the sure option. If the dollar values are positive, you win that amount. If the dollar values are negative, you lose that amount. Remember that you are a teenager and should act according to your preferences as a teenager. |

| Middle aged | Image that you are middle aged. You are given a prospect and a set of sure options. You will compare the prospect to each of the sure options one-by-one. If you reject the sure option, you would play the prospect. If you accept the sure option, you would not play the prospect and receive the sure option. If the dollar values are positive, you win that amount. If the dollar values are negative, you lose that amount. Remember that you are middle aged and should act according to your preferences as a middle aged person. |
|---|---|
| Senior citizen | Imagine that you are a senior citizen. You are given a prospect and a set of sure options. You will compare the prospect to each of the sure options one-by-one. If you reject the sure option, you would play the prospect. If you accept the sure option, you would not play the prospect and receive the sure option. If the dollar values are positive, you win that amount. If the dollar values are negative, you lose that amount. Remember that you are a senior citizen and should act according to your preferences as a senior citizen. |

**Table 13:** System prompts for role-playing.

| | Prompt |
|---|---|
| Teenager | Imagine that you are giving advice to a teenager. They are given a prospect and a set of sure options. They will compare the prospect to each of the sure options one-by-one. If they reject the sure option, they would play the prospect. If they accept the sure option, they would not play the prospect and receive the sure option. If the dollar values are positive, they win that amount. If the dollar values are negative, they lose that amount. Remember that you are giving advice to a teenager and should give advice according to their preferences as a teenager. |
| Middle aged | Imagine that you are giving advice to someone middle aged. They are given a prospect and a set of sure options. They will compare the prospect to each of the sure options one-by-one. If they reject the sure option, they would play the prospect. If they accept the sure option, they would not play the prospect and receive the sure option. If the dollar values are positive, they win that amount. If the dollar values are negative, they lose that amount. Remember that you are giving advice to a middle aged person and should give advice according to their preferences as a middle aged person. |

| | |
|---|---|
| Senior citizen | Imagine that you are giving advice to a senior citizen. They are given a prospect and a set of sure options. They will compare the prospect to each of the sure options one-by-one. If they reject the sure option, they would play the prospect. If they accept the sure option, they would not play the prospect and receive the sure option. If the dollar values are positive, they win that amount. If the dollar values are negative, they lose that amount. Remember that you are giving advice to a senior citizen and should give advice according to their preferences as a senior citizen. |

Table 14: System prompts for giving advice.

