# OpenReview forum: "LLM economicus? Mapping the Behavioral Biases of LLMs via Utility Theory"
_colmweb.org/COLM/2024/Conference — COLM_

### Official Review · Reviewer_Rvzo · 2024-05-10

**Rating:** 6
**Confidence:** 3
**Ethics Flag:** 1

**Summary:**

This paper uses the set up of common behavioral economic games to attempt to measure four properties of LLMs: inequity aversion, risk aversion, loss aversion, and time discounting. That is, using a standard prompting and decoding setup, the authors ask models to make decisions about things like how much money to offer another a player (or whether to accept such an offer) in various hypothetical scenarios. As a precursor to this step, the authors attempt to assess the models' "competence", and then only test a subset of the models that pass this first step. The authors demonstrate that for at least some of these agent properties, such as risk aversion, a subset of models end up falling between typical (?) human behavior and "perfectly rational" behavior, whereas in others they differ from both.

**Reasons To Accept:**

Although this paper doesn't make the case very strongly, the question of how LLMs will behave in various decision-making scenarios that test rationality is clearly important for many hypothetical downstream applications.

This paper makes what seems like a reasonable first pass attempt at this area, using a combination of well-established behavior economic games, and testing a wide variety of models.

The authors also include some experiments in which they try to modify the behavioral characteristics of models via prompting, which is a natural extension of the first part of the paper.

**Reasons To Reject:**

Although this paper is interesting, there are a few issues with it, and my sense is that it could probably benefit from another round of revisions.

The most basic issue is the relatively lightness of the experiments. As far as I can tell, the authors just tested a single overall prompt design, and no justification is given for why this particular wording was used, or how sensitive these results might be to minor variations (of which there are innumerable possibilities).

In addition, the authors' use of a "competence" test seems strange, given that this seems to be yet another basically single prompt setup that may or may not relate to the substance of the game. (It clearly does if we assume that models are understanding these texts like people do, but that is not a well justified assumption). Relying on such a narrow assessment to filter out models seems odd, both when it's not clear why models need to be filtered out, but also when there are so many benchmarks on performance that might be used as proxies.

A second issue has to do with presentation. Although the text is clear, the results are not presented in a way that has been streamlined for legibility to an audience that may or may not be familiar with these economic games. For example, Figure 2 contains a table with three columns that can only be interpreted by finding the meanings of symbols in the text; similarly for Figure 3, with no attempt to provide any interpretation of the figures for the reader in the captions. Much of the text in the figures is too small to be able to read clearly, and some of the colors used are hard to distinguish. In Figure 1, the response of "I offer $4" seems like a mismatch from the actual experimental setup, which has the potential to mislead people about how precisely the models are engaging in these games.

Third, I might have missed this, but it's not clear to me where the "Human" results are coming from? Perhaps these are purely theoretical or based on some sort of existing model, but it doesn't seem like the authors ran new experiments with people using identical setups as for the models, making me wonder if they are directly comparable.

Fourth, although the authors make a minimal attempt to explore the idea of personas or role-playing for models, much more could be done in this direction. For example, do we see patterns of responses that are consistent with real people as you change demographic characteristics such as age (to use one example the authors highlight).

Finally, some of the language in this paper is somewhat poorly chosen. The idea that the authors' competence test demonstrates that the a model "understands" the prompt seems like a bit of a leap. (By contrast, the authors don't grapple with the idea that models might understand the setting too well; that is, what is "rational" for a LLM playing a hypothetical game might differ from that of a person). The authors final comment about the "fabled relationship between the expert system ELIZA and the author's assistant" is a bit tone deaf, especially given Weizenbaum's extensive future thoughts about this topic.

---

> ### Author Rebuttal · Authors · 2024-05-30
>
> Thank you for your thoughtful review.
>
> **Weakness 1 (experimental lightness)**. We would like to push back on this point, as we performed prompt ablations (Appendix B.3, C.3, D.3) on the system and user prompts. For most ablations, we found that the formulation of the prompts had little effect on the strategic choice given repeated samples of the learned distribution. We attempted as much experimental rigor as possible within a limited research budget of compute and API credits. We will make this point clearer in the next iteration of the paper.
>
> **Weakness 2 (competence tests)**. The competence tests are designed to test specific competencies related to playing each game. We assume that we cannot meaningfully analyze an LLM’s behavior if the LLM cannot execute the basic reasoning capabilities to play the game. (We also tried various prompt ablations to ensure that the competence tests were robust). Indeed, our tests are meaningful only if we (at least weakly) assume that LLMs “are understanding these texts like people do”.  While this is a valid critique, this assumption is implicit in pretty much every prompt-based interpretability work, which is one of the only means by which we can probe the most capable LLMs.
>
> **Weakness 3 (presentation)**. We will explicitly explain variables in the accompanying captions for each figure and update plots to be differentiated by pattern in addition to color.
>
> **Weakness 4 (human results)**. Human results are taken from (classic) studies in experimental economics; we did not run our own human studies. We report parameters from papers cited in each game (Fehr & Schmidt 1999, Kahneman & Tversky 1992, Rachlin et al 1991 respectively); we will clarify how we compare human behavior in Section 3 and explicitly cite each value in figure footnotes. We believe the human results are comparable because we directly adapt the experiments reported in these papers to LLMs.
>
> **Weakness 5 (personas)**. Section 4 was intended as a case study, and we intend to further examine the effect of personas and role-playing in follow-on work. The main contribution of our paper is to show that utility theory provides interesting tools for probing the economic behavior of LLMs. Whether and how LLM behavior can be changed through prompting (e.g., roles-based prompting) or finetuning is an interesting avenue for future work.
>
> **Weakness 6 (language choice)**. “Understand” and the ELIZA anecdote has unintended implications and will update accordingly.

---

> > ### Author Response · Authors · 2024-06-05
> > **Kind reminder of our response**
> >
> > Thank you again for your feedback! We hope that we have addressed your concerns. Since there's little time remaining for the discussion period, we would love to know if you have any lingering concerns that we can address within the timeframe.

---

> > ### Comment · Reviewer_Rvzo · 2024-06-06
> > **acknowledgement**
> >
> > Thank you for your response. I had not carefully read the appendices, so your pointer to the additional information is helpful, and I have raised my score by one point as a result, and look forward to the revised version.

---

### Official Review · Reviewer_hjkh · 2024-05-11

**Rating:** 6
**Confidence:** 3
**Ethics Flag:** 1

**Summary:**

The paper presents an LLM evaluation study of LLM’s economic decision-making abilities through utility theory. In particular, LLMs generally exhibit stronger inequity aversion, stronger loss aversion, weaker risk aversion, and stronger time discounting compared to human subjects.

**Questions To Authors:**

Can you provide clarification to the weaknesses mentioned above?

**Reasons To Accept:**

1. The study is novel and interesting
2. The paper is well-written and clearly organized.
3. The experiment results are interesting.

**Reasons To Reject:**

1. The technical depth of the paper is not enough. The paper simply presents LLM evaluation without (1) developing methods to understand what impacts LLM’s economic decision-making abilities or (2) teaching LLM to make rational economic decisions.

2. It would be beneficial for the reader if the author could clarify the association between LLM’s mathematical reasoning ability and economic decision-making abilities. This will help to understand the research's focus and implications more clearly.

---

> ### Author Rebuttal · Authors · 2024-05-30
>
> Thank you for your thoughtful review.
>
> **Weakness 1 (technical depth)**. We present a novel set of benchmarks and evaluation protocols - a topic of interest for COLM. We claim that our methods enable further studies into what impacts economic decision-making abilities, as shown in Section 4. For example, we show how our evaluation protocols can be used to study how implicit assumptions like age group affect an LLM’s risk and loss aversion, enabling researchers to probe for proxies that impact economic decisions. We focused on prompting techniques as the most economically interesting/competent models (i.e., GPT-4) were only accessible via prompting. However, we believe that the framework we present could be used to provide feedback to LLMs during fine-tuning, and such a technique could become practically relevant as more capable open-source models are released.
>
> **Weakness 2 (reasoning and decision-making)**.  Mathematical reasoning is closely related to rationality of economic decision-making insofar as it provides a set of tools necessary to analyze and model economic decisions. For example, the decision for whether or not to take an economic gamble (on a stock, insurance plan, or new home) relies on mathematical reasoning to calculate the expected value over time. However, human economic decision making is not rational in many cases, and some of the economic experiments that we consider in the study were originally designed to surface this irrationality.

---

> > ### Author Response · Authors · 2024-06-05
> > **Kind reminder of our response**
> >
> > Thank you again for your feedback! We hope that we have addressed your concerns. Since there's little time remaining for the discussion period, we would love to know if you have any lingering concerns that we can address within the timeframe.

---

> > ### Comment · Reviewer_hjkh · 2024-06-05
> > **Response to Authors**
> >
> > Thank you for your explanation and clarification. The reasoning and decision-making are well explained. However, I would like to point out that GPT-4 also supports model fine-tuning via API. I am happy to keep my score.

---

> > > ### Author Response · Authors · 2024-06-06
> > > **Thank you for your response**
> > >
> > > Thank you for your response! The fine-tuning API was out of our budget, but we hope to explore it in the future.

---

### Official Review · Reviewer_zUgL · 2024-05-11

**Rating:** 7
**Confidence:** 4
**Ethics Flag:** 1

**Summary:**

The paper introduces a novel framework for systematically evaluating the economic decision-making abilities of large language models (LLMs) through the lens of utility theory- a theoretical framework from behavioral economics. The paper adapts classic experimental games like the ultimatum, gambling, and waiting games to derive utility functions that quantify biases such as inequity aversion, risk/loss aversion, and time discounting in LLMs. The paper further conducted competency tests and then experimented with models that passed the different competency tests for their economic decision-making abilities.

This paper is well-written; the research design is rigorous and well thought-out. A key strength of this paper is the systematic and rigorous approach of grounding the analysis in established approaches from experimental economics and connecting them to LLMs. That approach demonstrates domain knowledge, further lending credibility to the findings. The paper also experimented with a diverse set of LLMs, highlighting the strengths and limitations of each LLM and how the persona represented in the prompts could influence the results. Overall, the contributions of this paper are strong, and it produces new knowledge that would benefit members of the LLM and alignment community in training LLMs to make sound economic decisions.

**Questions To Authors:**

Why did some of LLMs pass or fail the competency tests? It would have been interesting to provide a rationale for this result.

**Reasons To Accept:**

This paper is well written, the research design is rigorous and well thought out. The limitations of the paper are also acknowledged. Above all, this paper will make a valuable contribution to the LLM modeling community around designing LLMs that make sound economic decisions.

**Reasons To Reject:**

I do not see any reason to reject this paper.

---

> ### Author Rebuttal · Authors · 2024-05-30
>
> Thank you for your thoughtful review. We are glad you found our methods rigorous, comprehensive, and well-communicated. We likewise believe that this work presents a valuable set of tools for designing LLMs for economic decision-making, and we will be releasing an open-source package to ensure that others can easily adopt these tools.
>
> **Question 1 (competency tests)**. Thank you for your question, and we are happy to provide further rationale for the competence test for each game presented in the paper. For the ultimatum game, some LLMs did not pass the competence test because they were unable to do basic arithmetic (i.e. 10 dollars - 6 dollars = 4 dollars). For the gambling game, some LLMs did not pass the competence test because they were logically inconsistent about the gambles they take. For example, an LLM would be incompetent if the LLM was willing to bet 50 dollars at 20% probability, unwilling to bet 60 dollars at 20% probability, and willing to bet 70 dollars at 20% probability. For the waiting game, some LLMs did not pass the competence test because they were logically inconsistent about how long they were willing to wait for a monetary gain. For example, an LLM would be incompetent if the LLM was willing to wait 5 years for 50 dollars, unwilling to wait 10 years for 50 dollars, but willing to wait 25 years for 50 dollars. We will provide more examples of these failure modes and discuss this further in the next iteration of the paper.

---

> > ### Comment · Reviewer_zUgL · 2024-06-04
> >
> > Thanks for the additional clarification. I decided to keep my score as is and wish the authors all the best.

---

### Official Review · Reviewer_NcHx · 2024-05-15

**Rating:** 7
**Confidence:** 3
**Ethics Flag:** 1

**Summary:**

This paper describes a series of behavioral economic experiments adapted to be run with various LLMs rather than human subjects. The authors do this to probe whether the LLMs tend to show systemic behavioral biases in their decision behavior that is similar to or varies from the sorts of behavioral biases that are well-described in human economic decisions -- inequity aversion, risk and loss aversion, as well as time discounting.

The authors general find that while LLMs show non-rational biases of various kinds the manner and degree of these may differ from (at least their chosen comparison class) of human responses.

**Reasons To Accept:**

This is an, as far as I know, novel application of behavioral economic / decision making studies to LLMs. The use of utility function fitting as a formal modeling exercise allows the authors to quantify the ways in which LLM outputs systemically differ (or not) from human responses in the same or similar settings.

**Reasons To Reject:**

I'm not sure exactly what this finding tells us?

"the economic behavior of LLMs" -- this premise doesn't quite work.... when we ask people for judgements in these experiments those are assumed to reflect behavior that people would actually take. The LLMs have no grounding in the physical world and so it's not clear if the "behaviors" they would undergo / action they'd take is the the same as the text responses if a given LLMs were augmented with a grounding or linking function to the physical world

The categorical cutoff for the competence test -- do the models vary in their behavioral biases as a function of general "competence" here or on other evaluation measures? Otherwise hard to make a general statement about LLMs as a class.

---

> ### Author Rebuttal · Authors · 2024-05-30
>
> Thank you for your thoughtful review. We are glad you found our work novel and useful in comparing LLM behavior to human behavior. We believe that the ability to leverage existing human studies to quantifiably compare LLM behavior is a core strength of our approach.
>
> **Weakness 1 (premise)**. The reviewer suggests that because LLMs do not have grounding in the physical world, what they “say” may be different than what LLMs “do”, and hence probing the economic “behavior” of LLMs doesn’t really make sense. We agree that this is a valid (and deep!) point, and acknowledged some of these limitations in Section 5. However, we note that there is a rich body of work on treating LLMs as human-like agents in the context of economic/behavioral experiments (e.g., https://arxiv.org/abs/2208.10264  https://www.nber.org/papers/w31122 ), and these (influential) studies have yielded interesting findings and have spurred on much follow-up work. Moreover, we observe that even in the case of humans, in some of the experiments the questions are posed only as hypotheticals; we trust that the answers given by the experimental subjects reflect what they would truly do. From this point of view, we think that our experimental setup (as well as the experimental setup of the aforementioned studies) is not invalid. We nonetheless acknowledge that this point needs to be discussed more thoroughly, and will do so in the next iteration of the paper.
>
> **Weakness 2 (competence test)**. This is a good question. We do not see obvious patterns in the amount of bias as a function of their competence. To clarify further, the competence tests are designed to test specific competencies related to playing each game. For example, the competence test for the ultimatum game tests whether the LLM can do the requisite arithmetic to make an offer. We assume that we cannot meaningfully analyze an LLM’s behavior if the LLM cannot execute the basic reasoning capabilities to play the game. Given this, we agree that it may be hard to make general economic statements about LLMs as a class, or even as a single LLM family, given that an LLM may show competence in one experiment but not in another. However, once the LLM has passed the competence test, we can identify specific behavioral biases in specific versions of LLMs that may have profound impact on deploying these models in decision-making settings. We will add more discussion around this point.

---

> > ### Author Response · Authors · 2024-06-05
> > **Kind reminder of our response**
> >
> > Thank you again for your feedback! We hope that we have addressed your concerns. Since there's little time remaining for the discussion period, we would love to know if you have any lingering concerns that we can address within the timeframe.

---

> > > ### Comment · Area_Chair_2B8w · 2024-06-06
> > > **Reviewer please respond to this asap**
> > >
> > > A substantive rebuttal has been written; would the reviewer please respond before the approaching deadline.

---

> > > > ### Comment · Reviewer_NcHx · 2024-06-06
> > > >
> > > > Thanks for the thorough response. I liked the paper before (recommending it's acceptance) and I still do. My score remains unchanged.

---

### Decision · Program_Chairs · 2024-07-10

**Decision:**

Accept

**Comment:**

This paper is part of a growing body of work that treats LLMs as experimental subjects equivalent to humans, and tests their 'cognitive' biases and behaviours in much the same way as existing work in behavioural economics tests that of humans.

I think there are significant methodological questions about this whole research agenda, because it presupposes that LLMs have some kind of stable 'persona' that is being evaluated through the test, rather than being able to simulate different personae depending on how they are prompted and finetuned. Any lessons learned about their behaviour are really lessons about the default 'persona' induced through post-training by the LLM developers at a particular time, and tell us very little about how the LLM would behave in any practical setting, on the assumption that in any practical setting the user is likely to customise the model's persona, and of course the LLM developers themselves may be assumed to continue intervening in various ways on the default persona.

In an important sense, then, I actually disagree pretty robustly with the entire premise of the paper! However, reviewers etc far too often take that to be a sufficient reason to reject—it isn't! Though I think the research agenda is misguided, the paper is an excellent articulation of that research agenda: it is, as the reviewers agree, well-written, empirically robust, and informative. My criticisms aren’t a reason to reject the paper, but rather a reason for the authors to reflect further on precisely what it is they take themselves to be showing, and why it matters, as well as grounds for other people to write other papers that ask those methodological questions more directly, critically evaluating this research agenda accordingly. in other words, it's a reason for the authors to sit down with some philosophers of social science and have a good chat about the whole project. Which hopefully they'll be able to do at COLM.